# Adding *MASP1* to the lectin pathway—Leprosy association puzzle: Hints from gene polymorphisms and protein levels

**Hellen Weinschutz Mendes**[1¤]*, **Angelica Beate Winter Boldt**[1,2], **Ewalda von Rosen Seeling Stahlke**[3], **Jens Christian Jensenius**[4], **Steffen Thiel**[4], **Iara J. Taborda Messias-Reason**[1]

**1** Laboratory of Molecular Immunopathology, Department of Clinical Pathology, Clinical Hospital, Federal University of Paraná, Curitiba, Brazil, **2** Laboratory of Human Molecular Genetics, Department of Genetics, Federal University of Paraná, Curitiba, Brazil, **3** Health State Department of Paraná, Curitiba, Brazil, **4** Department of Biomedicine, Aarhus University, Aarhus, Denmark

¤ Current address: Department of Biology, University of Ottawa, Ottawa, Ontario, Canada
* hellen.chrisw@gmail.com

**Data Availability Statement:** All relevant data are included within the paper and its Supporting Files.

## Abstract

### Background

Deposition of complement factors on *Mycobacterium leprae* may enhance phagocytosis. Such deposition may occur through the lectin pathway of complement. Three proteins of the lectin pathway are produced from the gene *MASP1*: Mannan-binding lectin-associated serine protease 1 (MASP-1) and MASP-3 and mannan-binding lectin-associated protein of 44 kDa (MAp44). Despite their obvious importance, the roles played by these proteins have never been investigated in leprosy disease.

### Methodology

We haplotyped five *MASP1* polymorphisms by multiplex sequence-specific PCR (intronic *rs7609662*G>A* and *rs13064994*C>T*, exon 12 3'-untranslated *rs72549262*C>G*, *rs1109452*C>T* and *rs850314*G>A*) and measured MASP-1, MASP-3 and MAp44 serum levels in 196 leprosy patients (60%, lepromatous) and 193 controls.

### Principal findings

Lower MASP-3 and MAp44 levels were observed in patients, compared with controls (P = 0.0002 and P<0.0001, respectively) and in lepromatous, compared with non-lepromatous patients (P = 0.008 and P = 0.002, respectively). Higher MASP-3 levels were present in controls carrying variants/haplotypes associated with leprosy resistance (*rs13064994*T*, *rs1109452_rs850314*CG* within *GT_CCG* and *rs850314*A*: OR = 0.5–0.6, Pcorr = 0.01–0.04). Controls with *rs1109452*T*, included in susceptibility haplotypes (*GT_GTG/ GT_CTG*: OR = 2.0, Pcorr = 0.03), had higher MASP-1 and lower MASP-3 levels (P≤0.009). Those with *GC_CCG*, presented increasing susceptibility (OR = 1.7, Pcorr = 0.006) and higher MAp44 levels (P = 0.015). MASP-3 expression decreased in patients,

The genotyping protocols can be found at: https://www.protocols.io/view/masp1-multiplex-pcr-ssp-rs7609662-rs13064994-rs725-27kghkw. The accession numbers and their associated databases are: NCBI: 5648, MIM: 600521, ENSEMBL: ENSG00000127241, hprd: 02749, UniProtKB: P48740.

**Funding:** This work was supported by the 01/2007 and 518/2010 PRODOC grants of CAPES (Coordenação de Aperfeiçoamento de Pessoal Superior, http://www.capes.gov.br/bolsas/bolsas-no-pais/prodoc) and by the 034/2008 CNPq (Conselho Nacional de Desenvolvimento Científico e Tecnológico, http://www.cnpq.br/web/guest/bolsas-e-auxilios) and Fundação Araucária (http://www.fappr.pr.gov.br/) grants for ABW Boldt and IJT Messias-Reason. Dr. Thiel and Dr. Jensenius were funded by the Danish Research Council and the Novo Nordisk Foundation. The funders had no role in study design, data collection and analysis, decision to publish, or preparation of the manuscript. Hellen Weinschutz Mendes had a MSc scholarship from Coordenação de Aperfeiçoamento de Pessoal Superior (CAPES).

**Competing interests:** The authors have declared that no competing interests exist.

compared with controls carrying *rs1109452_rs850314\*CA* or *CG* (P≤0.02), which may rely on exon 12 CpG methylation and/or miR-2861/miR-3181 mRNA binding.

## Conclusion

Polymorphisms regulating MASP-3/MAp44 availability in serum modulate leprosy susceptibility, underlining the importance of lectin pathway regulation against pathogens that exploit phagocytosis to parasitize host macrophages.

## Author summary

Since immemorial times, *Mycobacterium leprae* inflicts permanent injuries in human kind, within a wide symptomatic spectrum ranging from insensitive skin patches to disabling physical lesions. Innate resistance to this parasite is well recognized, but poorly understood. The complement system is one of the most important arms of the innate response, and several lines of evidence indicate that it may be usurped by the parasite to enhance its entrance into host cells. These include our recent work on genetic association of the disease with lectin pathway components and the complement receptor CR1, whose polymorphisms modulate susceptibility to infection and clinical presentation. Here, we add another pivotal piece in the leprosy parasite-host interaction puzzle: polymorphisms and serum levels of three different lectin pathway proteins, all encoded by the same gene, namely mannan-binding lectin-associated serine protease 1 (*MASP1*). We found lower levels of two of these proteins, MASP-3 and MAp44, in leprosy patients. Higher MASP-3/lower MASP-1 levels were associated with protective haplotypes, containing two side-by-side polymorphisms located in the exclusive untranslated region of MASP-3 exon 12, which may regulate exon splicing and/or translation efficiency. The associations revealed in this study reflect the pleiotropic nature of this gene. They further illustrate the complexity of the response mounted against the parasite, which places *MASP1* products in the regulatory crossroad between the innate and adaptive arms of the immunological system, modulating leprosy susceptibility.

## Introduction

Leprosy is a chronic infectious disease caused by the obligate intracellular bacteria *Mycobacterium leprae*, with a 2018 global prevalence rate of 0.2/10,000 population, irreversibly disabling about 5.5% of 210,000 new cases every year. Brazil contributed to 92.3% of the increase in new American cases from 2017–18 [1]. Clinical manifestations depend on the host's genetic polymorphisms and environmental factors that modulate the quality of the immune response, ranging from the multibacillary disabling lepromatous from one end of the clinical spectrum to the paucibacillary tuberculoid form at the other end, with borderline forms in between [2]. The very ancient origin of the complement system [3], made it an ideal platform to coevolve with pathogens, like *M. leprae.* To assist infection, *M. leprae* bacteria usurp complement activation to be opsonized and more readily phagocytosed into macrophages, one of their preferred host cells. In fact, the lectin pathway of complement (LP) has long been recognized as favoring the establishment of infection and development of disease [4–10]. There is also strong evidence that intracellular C3 cleavage directs T cell activation towards the Th1 pole, which is associated with the paucibacillary presentation of the disease. Since the first suggestion of balancing

selection operating on the polymorphism of the gene encoding mannose-binding lectin (*MBL2*) due to protection against leprosy [4], much has been done investigating the possible roles played by LP genes and their products on the susceptibility to this disease [5–9] [11–15].

The lectin pathway of complement starts with the recognition of pathogen- or damaged/ altered cell-associated patterns of carbohydrates or patterns of acetylated groups by pattern recognition molecules (PRMs i.e. collectins, MBL and ficolins—FCNs). These PRMs form circulating complexes with homodimers of MBL-associated serine proteases (MASPs) or MBL-associated proteins (MAps). Upon collectin/ficolin binding to a target, MASP-1 autoactivates and transactivates MASP-2, leading to the cleavage of complement factors C2 and C4, in order to form the C3 convertase (C4bC2b), also produced by the classical pathway. C4bC2b cleaves C3 and generates the anaphylatoxin C3a and the opsonin C3b. C3b is deposited on the target and recognized by complement receptors (CRs), which internalize the opsonized elements into phagocytes. Alternatively, formation of the C5 convertase (C4bC2bC3b) generates membrane pores (membrane attack complexes) formed by C5b and C6, C7, C8 and C9. The C3a and C5a anaphylatoxins further attract immune cells to the site of activation [16,17].

The *MASP1* gene (3q27.3) encodes the serine proteases MASP-1 and MASP-3 and the non-enzymatic protein MAp44 (or MAP-1) [18]. These three proteins circulate in plasma as homodimers complexed with PRMs. The substrate specificity of MASP-1 is quite broad, resembling thrombin and trypsin. Besides cleaving MASP-2 in the lectin pathway, MASP-1 has procoagulant activity, cleaving and activating Factor XIII and fibrinopeptide, generating fibrinopeptide B and attracting neutrophils to assist the coagulation cascade [19]. It also activates carboxypeptidase B2, a molecule that prevents fibrinolysis and inactivates C3a and C5a anaphylatoxins [20]. MASP-1 generates bradykinin from the cleavage of high-molecular-weight kininogen [21]. It also cleaves PAR4 (protease-activated receptor 4) on endothelial cells and induces MAPKp38 (mitogen activated protein kinase protein 38) and NFkB (nuclear factor kappa-light-chain-enhancer of activated B cells) proinflammatory signaling (reviewed by [22] and [17]).

The alternative pathway, which is at least as old as the lectin pathway, becomes activated when MASP-3 cleaves pro-Factor D, enabling factor D to create the alternative pathway C3 convertase. In the absence of MASP-3, only thrombin possibly cleaves pro Factor D, but under circumstances of ongoing coagulation [23]. MASP-3 and Map44 also compete with MASP-1 and MASP-2 for the same binding sites on the PRMs, thus inhibiting activation of the lectin pathway [24]. MAp44, highly expressed in the heart [25], also has the ability of displacing MASPs from within the collagenous stalks of the PRMs, which may reduce myocardial tissue damage after ischemia-reperfusion injury [26–28]. Rare mutations in a highly conserved region of exon 12 of the *MASP1* gene, which is exclusive of MASP-3 and encodes the serine protease domain of this protein, cause the 3MC1 (Malpuech-Michels-Mingarelli-Carnevale) syndrome, pointing to an important role in ectodermal development [29].

The possible roles of MASPs in the establishment of infections and in leprosy progression are still poorly understood. Low MASP-2 levels, as well as *MASP2* polymorphisms associated with low MASP-2 production, were associated with increased susceptibility to leprosy [8]. Low MBL levels and corresponding *MBL2* polymorphisms, in contrast, were associated with increased resistance [11,12], and higher FCN-3 levels were more frequent in leprosy patients than in controls [14]. It has also been suggested that complement receptor CR1 and CD91/calreticulin bind the collagenous chains of collectins and ficolins deposited on pathogens or altered cells, leading to their internalization, but that MASPs and MAps compete with this binding site, preventing this recognition [30,31]. CR1 binds opsonized *M. leprae* to enter the cell [32], and may use C3b and collectins/ficolins. Interestingly, we recently found polymorphisms of the *CR1* gene associated with leprosy, as well as a negative correlation between the

anti-inflammatory soluble CR1 and pro-inflammatory MBL levels, probably preventing inflammation [15]. All these associations were found with South Brazilian leprosy patients.

Given this context, we investigated whether *MASP1* gene variants and products are associated with susceptibility to leprosy and to the different clinical forms of the disease. We aim to provide a better understanding of the immunological clinical spectrum of leprosy and of the role played by the lectin pathway in mycobacterial infections.

## Materials and methods

### Ethics statement

This case-control, cross-sectional study was conducted according to the Declaration of Helsinki. The local medical ethics committee of the HC-UFPR approved the study (protocol 497.079/2002–06, 218.104 and 279.970). All subjects were adults and signed a written informed consent.

### Subjects and samples

We included 196 leprosy patients, of which 138 were consecutive outpatients from the Clinical Hospital of the Federal University of Paraná (HC-UFPR) and 58, inpatients from the Sanitary and Dermatologic Hospital of Paraná, both in Curitiba, Brazil. Brazilian health workers are required to enter the clinical characteristics of each leprosy patient into the "SINAN" database (Sistema de Informação de Agravos de Notificação). Each patient file contains information regarding: (1) number and location of cutaneous lesions and affected neural trunks, (2) degree of physical disability with involvement of the eyes, hands and feet, (3) bacilloscopic exam (acid-resistant bacilli screening in serous skin) and (4) histopathology results. Based on the registered clinical and histopathological features, patients were diagnosed and classified according to Ridley and Jopling criteria [33]. As the lepromatous form of leprosy is not only the most disabling, but also the most prevalent in our cohort, we sought to carry out comparisons between this patient group and the less-affected, non-lepromatous individuals. For association analyses regarding this clinical presentation, the lepromatous group comprised only patients presenting the lepromatous form of leprosy, while the non-lepromatous group comprised patients classified in all the other forms, namely tuberculoid, borderline and intermediate (those non-specified were not included). The power of the allele and haplotype analysis within the patient group was too low for detecting small effects, and results shall be taken with caution. On the other hand, within this same group, the power of the comparison of protein serum levels reached 70%, increasing our confidence in the results.

The control group comprised 214 blood donors with the same socioeconomic, ethnic and geographic background as the patients, according to information from the Hemepar and HC-UFPR blood banks. Patients and controls were defined as Euro-, Afro-Brazilians or Amerindians, based on physical characteristics and ancestry information based on the origin of first-degree relatives, collected upon patient consultations or from blood bank files. This actually means 9% sub-Saharan African and 5% average Amerindian genetic component for Euro-Brazilians, and at least 40% of African and 6% of Amerindian ancestry for Afro-Brazilians, based on the HLA allele distribution of South Brazilian populations, formerly classified in the same way. Amerindians present a very low admixture degree with European and African populations [34,35] (Table 1). Blood was collected with, or without anticoagulant ethylenediaminetetraacetic acid (EDTA) for serum collection, and DNA was extracted from peripheral blood mononuclear cells through commercial kits (Qiagen GmbH, Hilden, Germany and GFX Genomic Blood DNA Purification Kit, GE Healthcare, São Paulo, Brazil).

**Table 1. Clinical and demographic description of controls and leprosy patients.**

| Parameters | Controls | Patients | Exact P value |
|---|---|---|---|
| N<br>Age average [Min-Max]<br>Male (%) | 214<br>38.17 [18–61]<br>116 (54.2) | 196<br>51.31 [18–94]<br>119 (60.7) | -<br><0.0001<br>0.195 |
| Ethnical background (%)* | | | 0.70 |
| Euro-Brazilian | 176 (82.2) | 158 (80.6) | - |
| Afro-descendant<br>Amerindian | 34 (17.3)<br>4 (2.1) | 36 (18.6)<br>2 (1.1) | -<br>- |
| Clinical Form (%) | | | |
| Lepromatous | n.a. | 118 (60.2) | n.a. |
| Borderline | n.a. | 27 (13.7) | n.a. |
| Tuberculoid<br>Indeterminate<br>Non-specified | n.a.<br>n.a.<br>n.a. | 18 (9.2)<br>10 (5.1)<br>23 (11.7) | n.a.<br>n.a.<br>n.a. |

n: number of individuals; na.: not applicable

*: Ethnic background based on physical characteristics and ancestral information, corroborated by HLA genotyping of South-Brazilians classified in the same way (Probst et al. 2000, Braun-Prado et al. 2000).

## *MASP1* genotyping and haplotyping

A sequence-specific multiplex amplification method (multiplex PCR-SSP) was optimized in order to haplotype five single nucleotide polymorphisms (SNPs): *rs7609662\*G>A* and *rs13064994\*C>T* in intron 1 and *rs72549262\*G>C*, *rs1109452\*C>T* and *rs850314\*G>A* in exon 12 within the 3' untranslated (UTR) region (reference sequence: ENST00000337774.9). We amplified a 730 bp fragment specific for rs7609662 and rs13064994 in intron 1 and co-amplified a 365 bp fragment specific for rs72549262 and rs1109452+rs850314 (both are adjacent SNPs) in exon 12, all in a batch of four low-cost reactions, as previously described for *MASP2* [8]. As a control for the amplification quality, we co-amplified a 500 bp fragment in every single reaction, corresponding to exon 8 of the Ficolin 2 gene (*FCN2*) by adding two generic primers (**Table 2**). The protocol starts with a denaturation step of 3 min at 96˚C, followed by 35 cycles of 20 sec at 94˚C for denaturation, 30 sec for primer annealing at variable temperatures (see below), and 30 sec DNA extension at 72C, concluding with 1 min and 30 sec at 72˚C or extension. We used three different annealing temperatures according to previously published "touch-down" protocol: the 10 first cycles at 61C, followed by 10 cycles at 59˚C and 15 cycles at 57˚C. The haplotypes defined by these five SNPs, were identified by the presence or absence of specific bands in agarose gel, after electrophoresis.

**Table 2. *MASP1* sequence-specific primers and fragment size.**

| Forward Primers | Reverse Primers | Fragment size |
|---|---|---|
| **Intron 01** | | |
| *MASP1* rs7609662_Af 5' ATATTTGTTTCATATGTTTGAAACC**A** 3' | *MASP1* rs13064994_Cr 5' TTCTTAAACCAATCTGTGGAA**G** 3' | 730 bp |
| *MASP1* rs7609662_Gf 5' ATATTTGTTTCATATGTTTGAAACC**G** 3' | *MASP1* rs13064994_Tr 5' TTCTTAAACCAATCTGTGGAA**A** 3' | 730 bp |
| **Exon 12** | | |
| *MASP1* rs72549262_Cf 5' CCCTCTCTCTTAGTGTGAT**C** 3' | *MASP1* rs1109452_Tr 5' CGACTAAGTCCCCATATTC**A** 3' | 365 bp |
| *MASP1* rs72549262_Gf 5' CCCTCTCTCTTAGTGTGAT**G** 3' | *MASP1* rs1109452_Cr 5' CGACTAAGTCCCCATATTC**G** 3'<br>*MASP1* rs850314_Ar 5' CGACTAAGTCCCCATATT**T** 3' | 365 bp<br>366 bp |

Each primer is named after the SNP it amplifies, f: forward; r: reverse. In bold: variant nucleotides; bp: base pairs.

Using haplotype phasing through sequence-specific amplification, we identified three intron 1 haplotypes: (*AC*, *GC*, *GT*) and four exon 12 haplotypes (*CCA*, *CCG*, *CTG* and *GTG*). We reconstructed the whole haplotypes using a Pseudo-Bayesian Algorithm for combining intron 1 and exon 12 haplotypes, implemented in the ARLEQUIN v.3.1 software. This approach resulted in a total of twelve *MASP1* haplotypes in leprosy patients and thirteen in controls. All these haplotypes were analyzed for possible associations with leprosy and with MASP-1, MASP-3 and MAp44 levels.

## MASP-3 and MAp44 levels assays

Serum concentrations of MASP-3 and MAp44 were determined by time-resolved immuno-fluorimetric assays (TRIFMA) for 142 and 145 patients, respectively, and 116 controls, as previously described [27]. Briefly, samples were diluted in binding buffer, 40-fold for MAp44 detection and 100-fold for MASP-3, and incubated in microtiter wells coated with a monoclonal antibody. The bound protein is detected by a specific biotin-labeled monoclonal antibody, which is then subsequently detected by europium-labeled streptavidin. The provided signal is measured by time-resolved fluorometry. Four internal controls were added to each assay plate in both assays.

## MASP-1 levels assay

The time-resolved immunofluorimetric assay for MASP-1 is an inhibition assay, where circulating MASP-1 in the sample inhibits the binding of an anti-MASP-1 antibody to a surface coated with a fragment of MASP-1, as previously described [36]. Briefly, diluted serum samples of 141 patients and 116 controls, 60-fold in binding buffer, were incubated with an equal volume of diluted rat anti-MASP-1 antibody for approximately an hour and then added to the coated microtiter wells. Bound rat anti-MASP-1 were detected with biotinylated rabbit anti-rat-Ig followed by europium-labeled streptavidin, where bound europium is measured by time-resolved fluorometry. Four internal controls were also added to each plate for this assay.

## Statistics

Genotype, allele and haplotype frequencies were obtained by direct counting. The expectation maximization (EM) algorithm was used to calculate maximum likelihood estimates of intron 1 −exon 12 haplotype frequencies, while taking into account phase ambiguity. The hypothesis of Hardy–Weinberg equilibrium and of homogeneity between allelic distributions (exact test of population differentiation of Raymond and Rousset) was also evaluated with the ARLEQUIN software package version 3.1 (http://cmpg.unibe.ch/software/arlequin3/). We used the Wilcoxon-signed rank to define significant threshold levels and compared the distribution of protein serum concentrations between the groups, using nonparametric Mann-Whitney/ Kruskal–Wallis tests (since their distribution did not pass Shapiro-Wilk normality test), with Graphpad Prism 5.01 (GraphPad Software, La Jolla, CA). The reduced model of multivariate logistic regression was used to adjust results for demographic factors; as age, sex (factors that might influence protein levels [37]) and ancestry, using STATA v.9.2 (Statacorp, TX, USA). The P values obtained with multiple comparisons in the association studies were corrected with the Benjamini-Hochberg method.

## Accession numbers/ID numbers for *MASP1* gene

NCBI: 5648, MIM: 600521, ENSEMBL: ENSG00000127241, hprd: 02749, UniProtKB: P48740.

## Results

Protein serum levels in Southern-Brazilian patients and controls were within the range reported for a Danish population [24,27]. We found strong evidences for an association between MASP-3 and MAp44 serum levels and leprosy. We also identified a genetic association between MASP-1 and MASP-3 serum levels and *MASP1* polymorphisms within haplotypes associated with increased resistance and susceptibility to leprosy. The results are described in detail below.

### MASP-3 and MAp44 levels are associated with leprosy per se and lepromatous leprosy

First, we assessed possible associations of MASP-1, MASP-3 and MAp44 levels with leprosy and the different clinical forms of the disease. Leprosy patients presented lower MASP-3 levels (median 4,488 [1,722–14,634] ng/mL), than controls (median 5,575 [2,149–12,579] ng/mL) (Mann-Whitney P<0.001). In fact, the frequency of individuals with more than 5,500 ng/mL circulating MASP-3 in serum (a threshold defined with the Wilcoxon signed rank test– P<0.0004) was higher among controls: 51.7% or 60/116, compared with 31.7% or 45/142 in patients, independently of age and sex distribution (logistic regression OR = 0.51 [95% CI = 0.28–0.92] P = 0.026) (S1 Fig). Additionally, MASP-3 levels were even lower in lepromatous patients, the multibacilary form of disease presenting an exacerbated Th2 immune response. In these severely affected, often disabled patients, the median of MASP-3 levels was 4,209 [1,722–11,244] ng/mL, compared with 5,334 [2,21–14,634] ng/mL in patients with the other clinical forms (Mann-Whitney P = 0.0083). Individuals with MASP-3 levels higher than 5,500 ng/mL were also much more frequent among non-lepromatous (50% or 18/36), compared with lepromatous patients (26.8% or 26/97), independently of age and sex distribution (logistic regression OR = 0.38 [95%CI = 0.16–0.90], P = 0.028).

Similar associations were observed when analyzing MAp44 levels, but with a more conspicuous difference. Leprosy patients also presented lower MAp44 levels (median 1,715 [719–4,843] ng/mL in patients vs. median 2,330 [1,140–4,927] ng/mL in controls; Mann-Whitney P<0.0001) (Fig 1). As in the case of MASP-3, individuals with MAp44 levels higher than 2,300 ng/mL (a threshold level defined with the Wilcoxon-signed rank test – P<0.0001) were much more frequent among controls: 50.9% or 59/116, compared with 22.1% or 32/145 in patients, independently of age and sex distribution (logistic regression OR = 0.26 [95%CI = 0.14–0.49] P<0.0001) (S1 Fig). This pattern was also followed by lepromatous, compared with non-lepromatous patients: MAp44 median 1,646 [719–4,843] ng/mL vs. median 1,995 [985–4,359] ng/mL, respectively (Mann-Whitney P = 0.0021) (Fig 2). Individuals with MAp44 levels higher than 2,300 ng/mL were also much more frequent among non lepromatous (36.1% or 13/36), compared with lepromatous patients (15.5% or 15/97), again independent of age and sex distribution (logistic regression OR = 0.34 [95%CI = 0.13–0.89], P = 0.023).

In contrast, MASP-1 levels did not differ between patients and controls (median 7,036 [2,350–14,109] ng/mL vs. 6,207 [2,521–16,624] ng/mL, respectively; Mann-Whitney P = 0.173) or among the lepromatous patients and those with the other clinical forms (Mann-Whitney P = 0.603) (Figs 1 and 2).

MAp44 levels correlated significantly, but weakly, with the other two serine proteases (MASP-1: R = 0.21 in patients, Spearman P<0.05; MASP-3: R = 0.05 in patients, R = 0.36 in controls, both with Spearman P<0.0001). No correlation was found between MASP-1 and MASP-3 levels or MAp44 and MASP-1 levels in controls (S1 Fig). These results were expected, according to former reports [24] [36].

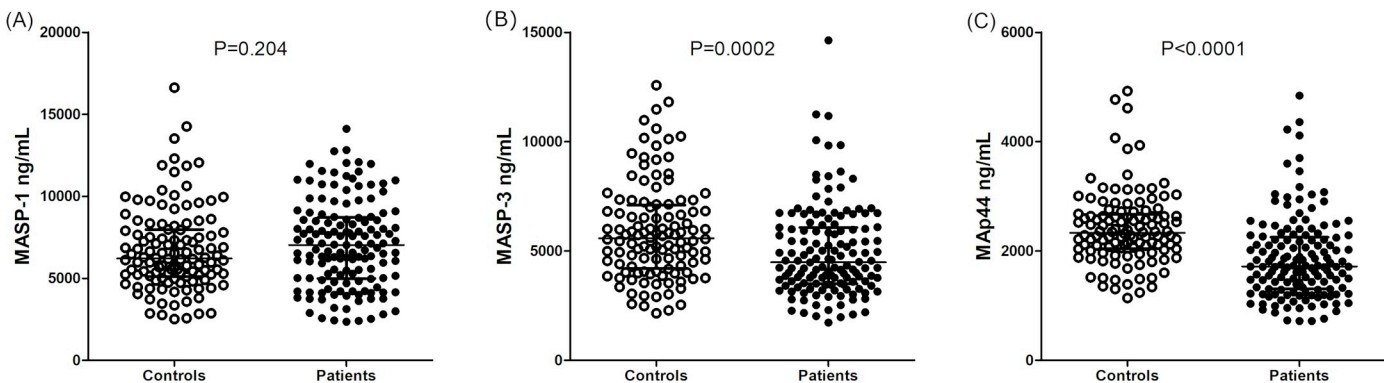

**Fig 1. MASP-1 (A), MASP-3 (B) and MAp44 (C) serum levels in controls and leprosy patients.** Data shown with medians and interquartile ranges and Mann-Whitney P values. Open and closed symbols represent controls and patients, respectively.

## MASP1 polymorphisms and haplotypes associated with leprosy

The allele frequencies for the investigated *MASP1* SNPs did not differ from Iberians (who contributed most to the Southern-Brazilian population), as well as from other Europeans, according to the 1000 Genomes project (exact test of population differentiation) (2014) (Table 3).

We carried out association analyses with the different haplotypes formed by intron 1 and exon 12 SNPs, as well as with all 5 SNPs combined. For intron 1 variants, we identified three haplotypes: *AC*, *GC* and *GT*. The *GC* combination accounted for more than half of all intron 1 haplotypes in the investigated groups. For exon 12 variants, we identified four haplotypes: *CCA*, *CCG*, *CTG* and *GTG*. Of these, *CCG* was the most common, but none of the others presented less than 5% frequency. The genotypic distributions of these haplotype combinations were in Hardy and Weinberg equilibrium, except for the distribution of exon 12 haplotypes in patients (P = 0.01). Analyses with these haplotypes revealed that distribution differed between leprosy patients and controls (exact test P = 0.016), as well as between lepromatous patients and controls (exact test P = 0.023), but not between lepromatous and non-lepromatous patients. In accordance with this absence of significant difference, there was no association of *MASP1* alleles/haplotypes/genotypes with the lepromatous clinical form of the disease, compared with the group containing the non-lepromatous forms. Furthermore, no associations with the disease (leprosy per se) occurred with the two variants located in intron 1. All other

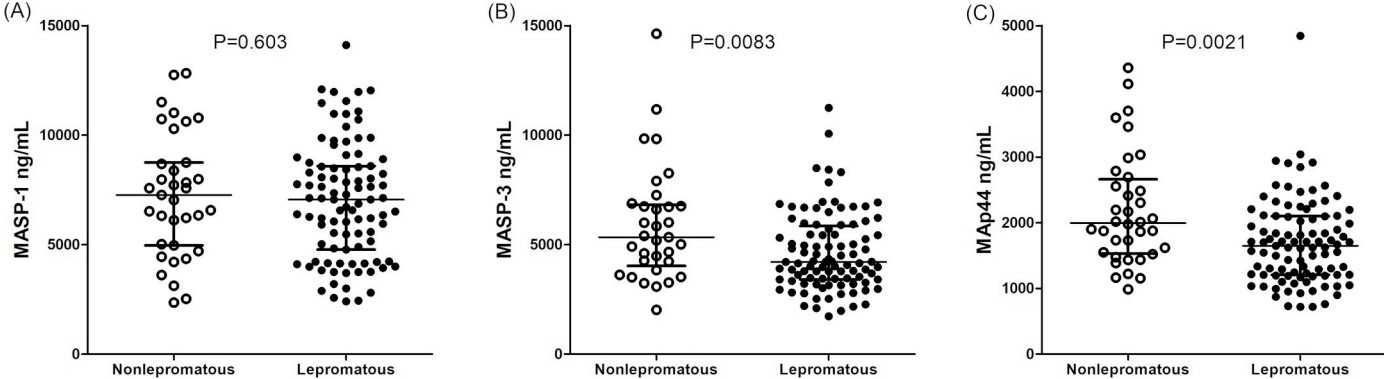

**Fig 2. MASP-1 (A), MASP-3 (B) and MAp44 (C) serum levels in non-lepromatous and lepromatous patients.** Data shown with medians and interquartile ranges and Mann-Whitney P values. Open and closed symbols represent controls and patients, respectively.

**Table 3. Association of *MASP1* variants and haplotypes with leprosy.** The intron 1 and exon 12 haplotypes were unambiguously build with sequence-specific amplification. The phase between them (symbolized by "_") was inferred using the expectation maximization algorithm. Official SNP nomenclature is given within parenthesis for the longest cDNA, corresponding to the mRNA transcript encoding MASP-1: ENST00000337774.9. Addit: Additive association model, which tests the hypothesis that homozygosity and heterozygosity for the minor allele are associated with leprosy (either with protection or with susceptibility), but homozygosity is stronger associated, than heterozygosity. Dom: Dominant association model, which tests the hypothesis that the carrier status of the minor allele (regardless if homozygous or heterozygous) is associated with leprosy (either with protection or with susceptibility). All associations were corrected for age, which was the only demographic factor that remained associated in the reduced model of logistic regression. *: GT_GTG + GT_CTG association. q*: Benjamini-Hochberg corrected p values; ns: not significant; OR: odds ratio; CI: confidence interval.

| Variants | Iberian % (n) | Controls % (n) | Patients % (n) | Lepromatous % (n) | Others % (n) | Model | Patients vs. Controls | |
|---|---|---|---|---|---|---|---|---|
| Total genotypes | 100 (107) | 100 (214) | 100 (196) | 100 (118) | 100 (55) | | | |
| rs7609662 (*c.5+2718G>A*) | | | | | | | OR [95% CI] | P (q*) value |
| A | 13.6 (29) | 14.1 (60) | 14.6 (58) | 13.5 (32) | 13.6 (15) | | ns | ns |
| G/G | 74.8 (80) | 74.3 (159) | 72 (141) | 72.8 (86) | 74.5 (41) | | ns | ns |
| G/A | 23.4 (25) | 23.3 (50) | 27 (53) | 27.2 (32) | 23.6 (13) | | ns | ns |
| A/A | 1.9 (2) | 2.3 (5) | 1 (2) | 0 (0) | 1.8 (1) | | ns | ns |
| rs13064994 (*c.6-2172C>T*) | | | | | | | | |
| T | 26.9 (49) | 28.5 (123) | 27.8 (109) | 30.1 (71) | 27.3 (30) | | ns | ns |
| C/C | 54.2 (58) | 50 (107) | 50.5 (99) | 45.7 (54) | 54.5 (30) | | ns | ns |
| C/T | 33.6 (36) | 43 (92) | 43.3 (85) | 48.3 (57) | 36.3 (20) | | ns | ns |
| T/T | 12.1 (13) | 7 (15) | 16.1 (12) | 3 (7) | 9.1 (5) | | ns | ns |
| rs72549262 (*c.1304-5229C>G*) | | | | | | | | |
| G | 8.9 (19) | 11.3 (48) | 7.7 (30) | 6.8 (16) | 5.4 (6) | | ns | ns |
| C/C | 82.2 (88) | 80.8 (173) | 87.2 (171) | 84.7 (100) | 91 (50) | | ns | ns |
| C/G | 17.8 (19) | 15.8 (34) | 10.2 (20) | 10.2 (12) | 7.3 (4) | | ns | ns |
| G/G | 0(0) | 3.2 (7) | 2.5 (5) | 1.7 (2) | 1.8 (1) | | ns | ns |
| rs1109452 (*c.1304-4903C>T*) | | | | | | | | |
| T | 25.2 (54) | 33.5 (143) | 33.7 (132) | 35.6 (84) | 30 (33) | | ns | ns |
| C/C | 57.9 (62) | 46.3 (99) | 44.9 (88) | 41.5 (49) | 51 (28) | | ns | ns |
| C/T | 33.6 (36) | 40.6 (87) | 42.8 (84) | 45.7 (54) | 38.2 (21) | | ns | ns |
| T/T | 8.4 (9) | 13.1 (28 | 12.2 (24) | 12.7 (15) | 10.9 (6) | | ns | ns |
| rs850314 (*c.1304-4902G>A*) | | | | | | | | |
| A | 32.7 (70) | 19.9 (86) | 15.3 (60) | 13.1 (31) | 18.1 (20) | | ns | ns |
| G/G | 47.7 (51) | 64 (137) | 73 (143) | 75.4 (89) | 71 (39) | | ns | ns |
| G/A | 39.3 (42) | 32.2 (69) | 23.4 (46) | 22.9 (27) | 21.8 (12) | Dom | 0.60 [0.37–0.96] | 0.035 (0.0438) |
| A/A | 13.1 (14) | 3.7 (8) | 3.5 (7) | 1.7 (2) | 7.3 (4) | | ns | ns |
| Intron 1_Exon 12 Haplotypes | | | | | | | | |
| GT_GTG * | | 0.7 (3) | 1.8 (7) | 1.7 (4) | 0.9 (1) | Addit | 2.19 [1.18–4.03] | 0.012 (0.025) |
| GT_CTG | | 6.1 (26) | 9.2 (36) | 10.6 (25) | 9.1 (10) | Dom | 2.01 [1.05–3.83] | 0.033 (0.03) |
| GT_CCG | | 16.6 (71) | 12.2 (48) | 12.7 (30) | 11.8 (13) | Dom | 0.52 [0.32–0.86] | 0.011 (0.0188) |
| GT_CCA | | 5.1 (22) | 4.6 (18) | 5.1 (12) | 5.4 (6) | | ns | ns |
| GC_GTG | | 49.3 (40) | 5.6 (22) | 7.2 (17) | 4.5 (5) | | ns | ns |
| GC_CTG | | 13.7 (59) | 14 (55) | 13.6 (32) | 12.7 (14) | | ns | ns |
| GC_CCA | | 14.0 (60) | 9.4 (37) | 6.7 (16) | 11.8 (13) | Dom | 0.48 [0.29–0.82] | 0.008 (0.0125) |
| GC_CCG | | 20.3 (87) | 28.6 (112) | 28.8 (68) | 30 (33) | Addit | 1.70 [1.21–2.40] | 0.002 (0.0063) |
| AC_CCA | | 2.3 (10) | 1.2 (5) | 1.2 (3) | 0.9 (1) | | ns | ns |
| AC_CTG | | 0.7 (3) | 2.8 (11) | 2.5 (6) | 1.8 (2) | | ns | ns |
| AC_GTG | | 1.1 (5) | 0.2 (1) | 0 (0) | 0.9 (1) | | ns | ns |
| AC_CCG | | 9.8 (42) | 10.2 (40) | 9.7 (23) | 10 (11) | | ns | ns |

associations were still significant after correction for multiple comparisons (q value) and for age (the only demographic factor that remained associated with the disease, in the reduced logistic regression model).

Linkage disequilibrium between the intron 1 and exon 12 alleles resulted in a total of twelve different *MASP1* haplotypes in leprosy patients and thirteen in controls, among which those with frequencies higher than 10% were *GC_CCG*, followed by *GT_CCG*, *GC_CTG*, *GC_CCA* and *AC_CCG*. Three of them were associated with leprosy, independently of any other demographic factor (**Table 3**).

Regarding the haplotypes formed by all variants analyzed in this work, the strongest association was found with the most frequent *GC_CCG* haplotype, which was associated with an additive (allele-dosage) susceptibility effect (OR = 1.70 [95%CI = 1.21–2.40], P<0.005). This is explained by a higher frequency of *GC_CCG* homozygotes and of *GC_CCG* heterozygotes among leprosy patients (21/196 or 10.71% and 70/196 or 35.71%), than among controls (16/214 or 7.48% and 55/214 or 25.7%), respectively. A dominant, age-dependent effect towards leprosy susceptibility was associated with carrying the less frequent *GT_CTG* haplotype (OR = 2.01 [95%CI = 1.06–3.83], P = 0.033). In other words, older individuals with this haplotype seem more prone to develop leprosy, if infected: there was 35/196 or 17.9% leprosy patients with *GT_CTG*, of which 26/35 or 74.3% with at least 40 years of age. In comparison, only 24/214 or 11.21% controls carried this haplotype, and only a third of them (8/24 or 33.3%) were 40 years of age or older. Nevertheless, the same analysis with either *GT_CTG* and/or another uncommon haplotype with *GT* in intron 1, namely *GT_GTG*, turned the association age-independent (OR = 2.19 [95%CI = 1.18–4.03], P = 0.012). Thus, age-dependency has a rather weak effect or may simply result from sampling bias.

In contrast, two haplotypes were associated with protection against leprosy. Among them, *GC_CCA* was associated with a dominant protective effect (OR = 0.48 [95%CI = 0.29–0.82], P = 0.008). This means that carriers of this haplotype were much more frequent among controls (59/214 or 27.6%), than among leprosy patients (36/196 or 18.4%). Similarly, controls presented a higher frequency of *GT_CCG* carriers (71/214 or 33.2%), compared with leprosy patients (43/196 or 21.9%). This haplotype was also associated with a dominant resistance effect against the disease (OR = 0.53 [95%CI = 0.32–0.86], P = 0.011) (**Table 3**).

### *MASP1* polymorphisms associated with protein serum levels

There was no association between the intron 1 *rs7609662*G>A* variant and *MASP1* protein concentrations. Yet, the neighboring intron 1 *rs13064994*C>T* polymorphism was associated with MASP-3 serum concentrations: healthy carriers with the *rs13064994*T* variant presented higher MASP-3 levels, than *C/C* homozygotes (medians 6,022 [2,286–11,820] ng/mL vs. 5,086 [2,149–12,580] ng/mL, respectively, P = 0.0103). This difference disappeared among leprosy patients, whose MASP-3 concentrations reached lower levels, independent of the genotype (medians 4,557 and 4,228 ng/mL, respectively) (Fig 3A).

Regarding the exon 12 variants, there was no association with the rs72549262 variant. However, controls with the minor *rs850314*A* allele of exon 12 presented higher MASP-3 levels, than *G/G* homozygotes (6,373 [2,286–11,820] ng/mL vs. 5,450 [2,149–11,480] ng/mL, P = 0.0342). This difference was not noticeable among patients, whose MASP-3 levels were generally lower (medians 4,500–4,554 ng/mL) and seemed no longer to be under the same genetic control (Fig 3B). In contrast, carriers of the minor *rs1109452*T* allele presented lower MASP-3 levels in controls, although they did not differ between healthy and diseased carriers (Fig 3C). Contrary to MASP-3 levels, MASP-1 serum concentration of *rs1109452*T* carriers were higher than in *C/C* homozygotes, independent of the disease (Fig 3D).

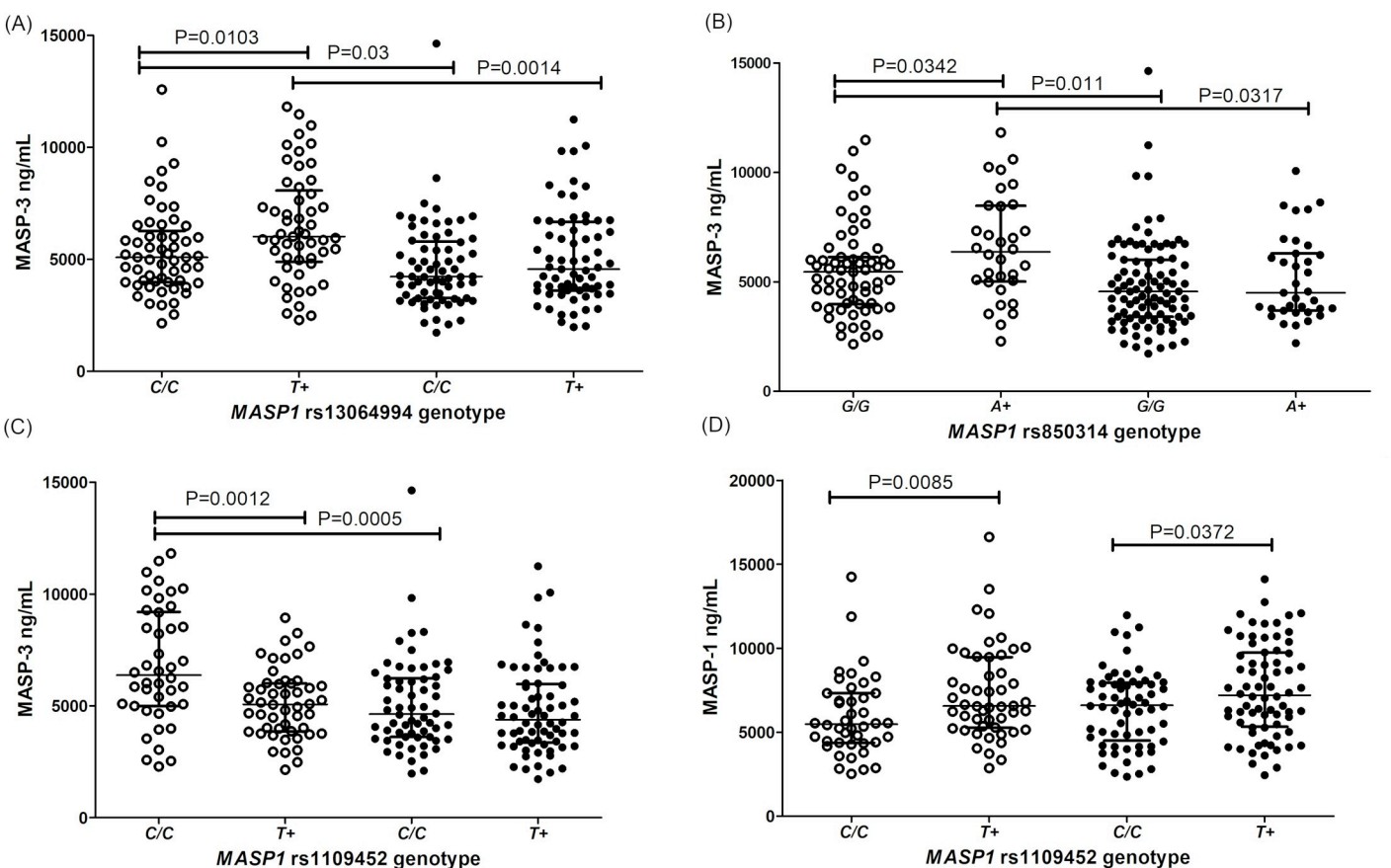

**Fig 3. Association between variant alleles and MASP levels. (A)** rs13064994 in intron 1 and MASP-3; **(B)** rs850314 in exon 12 and MASP-3; **(C)** rs1109452 in exon 12 and MASP-3; **(D)** rs1109452 in exon 12 and MASP-1. Data shown with medians and interquartile ranges and Mann-Whitney P values. Open and closed symbols represent controls and patients, respectively.

The adjacent exon 12 *rs1109452*C* and *rs850314*A*, as well as *rs1109452*T* and *rs850314*G* variants, occur in absolute linkage disequilibrium. The *CA*, *CG* and *TG* haplotype combinations did not present any association with MASP-1 and MAp44 levels (Fig 4A and 4C), although leprosy patients presented consistently lower MAp44 levels, regardless of the exon 12 genotype (Fig 4C). Healthy individuals with the *CA* or *CG* haplotype, presented higher MASP-3 concentrations than those with the *TG* haplotype (*CA* median 6,521–11,820] ng/mL and *CG* median 5,858 [2,286–11,820] ng/mL vs *TG* median 5,071 [2,149–8,941] ng/mL). In contrast to individuals with the *CA* and *CG* haplotypes, baseline levels of healthy individuals carrying *TG* do not differ from those with leprosy (Fig 4B).

Healthy individuals carrying the *GT_CCG* haplotype presented higher MASP-3 levels than those without it (median: 6,131 [2,286–11,820] ng/mL vs. 5,148 [2,149–12,580] ng/mL), a difference no longer noticed among leprosy patients (Fig 5A). Similarly, controls with the *GC_CCG* haplotype, but not patients, presented higher MAp44 levels (median 2,581 [1,355–4,927] ng/mL vs. 2,272 [1,140–4,068] ng/mL) (Fig 5B).

## Discussion

Parasitic mycobacteria are known to usurp and efficiently evade the host defense response (reviewed by [38] & [39]). However, investigating the immune response elicited by *M. leprae*

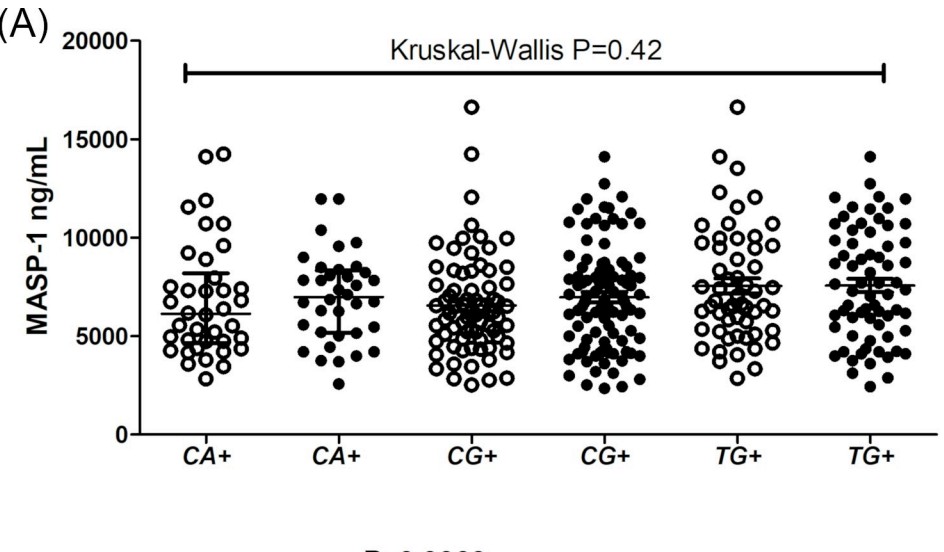

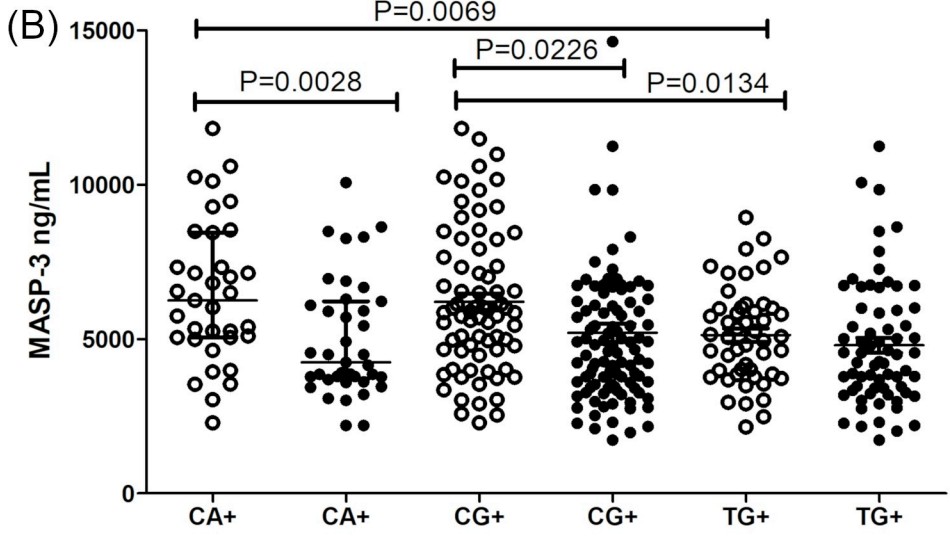

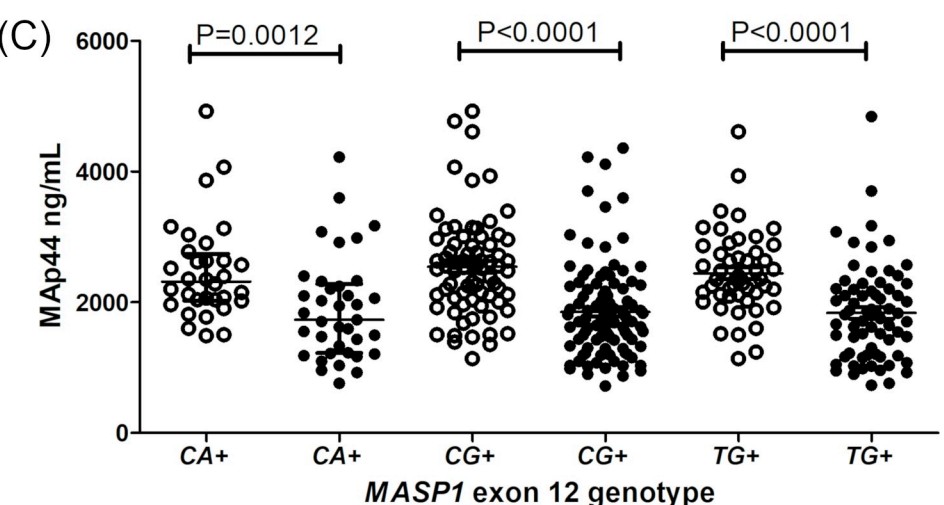

**Fig 4. Association between haplotypes with the rs1109452 and rs850314 adjacent exon 12 variants and levels of MASP1 products, MASP-1 (A), MASP-3 (B) and MAp44 (C).** Data shown with medians and interquartile ranges and Mann-Whitney P values. Open and closed symbols represent controls and patients, respectively. *CA*+: carriers of the *rs1109452\*C* and *rs850314\*A* variants. *CG*+: carriers of the *rs1109452\*C* and *rs850314\*G* variants. *TG*+: carriers of the *rs1109452\*T* and *rs850314\*G* variants. Unless if otherwise stated, comparisons were made with Mann-Whitney test.

remains a particular challenge, due to its extreme dependence on the human host. Genetic disease association studies shed light on a wide range of aspects from the onset of infection to disease classification, by uncovering genes whose protein products may play pivotal roles in this pathology [40]. This has been the case for several genes of the lectin pathway of complement; those encoding PRMs, *MBL2* [6,11] [7], *FCN1* [5], *FCN2* [9] and *FCN3* [14], the serine protease *MASP2* [8] and the MBL receptor encoded by *CR1* [15]. The evaluation of complement protein levels adds highly relevant information to this picture, as an indirect measure of gene expression, complement activation and consumption. Since the seventies, these measurements have been done for leprosy disease [41] [12,13], with results currently supported by transcriptome studies [42]. In the present investigation, we finally added *MASP1* polymorphisms and protein products, as one important piece of the initiation complexes of the lectin pathway to the association of complement with leprosy disease. In fact, we carried out the analyses presented in this work with the same patient cohort from a Brazilian population [5,9,11,14,15].

To understand the possible roles of *MASP1* products in the disease, it is important to keep in mind two prevailing hypotheses that may explain the role of complement proteins in leprosy disease. Firstly, they increase infection success by improving opsonization and phagocytosis of *M. leprae* by the host macrophage cells. Secondly, they increase inflammation after the disease is established, leading to more severe tissue damage.

Regarding the first hypothesis, it may be argued that any variant that reduces the rate of opsonin deposition would be protective, whereas any variant that increases opsonization would enhance susceptibility. According to this, one would expect that high MASP-1 levels would aid *M. leprae*'s entrance into host cells, whereas high MASP-3/MAp44 levels would block activation of the lectin pathway and reduce phagocytosis of the bacteria (although MASP-3 may also activate the alternative pathway). In fact, higher MASP-3 and MAp44 levels were characteristic for healthy individuals, although the expected effect was not seen for MASP-1 (Fig 6). With respect to the second hypothesis, it is expected that variants that reduce complement activation would (again) play a protective role. Indeed, we found a clear-cut difference between patients, with higher MASP-3/MAp44 levels more prevalent among those, less severely affected. Since it is known from former studies that Dapsone and Clofazimine treatment (used by the patients in this study) does not interfere with complement availability and function [43,44], it may be assumed that lower MASP-3 and MAp44 levels among patients, especially among those with the most severe lepromatous condition, are genetically determined (Fig 6).

There are numerous polymorphisms in the *MASP1* gene that may interfere with gene expression, some of which had been formerly investigated by others [45] [46]. We chose to investigate two SNPs located in a regulatory region of intron 1, which may interfere with the production of all three *MASP1* proteins, and three in exon 12, which is exclusive of MASP-3 and may uniquely affect the expression level of this protein. None of them had been previously investigated.

The *rs7609662\*A* in intron 1 is associated with higher *MASP1* mRNA levels in several tissues (https://gtexportal.org/home/snp/rs7609662), but we did not identify this effect at the protein level. The *rs13064994\*T* had the opposite effect (https://gtexportal.org/home/snp/rs13064994) on *MASP1* mRNA expression. We found an association of this allele with higher

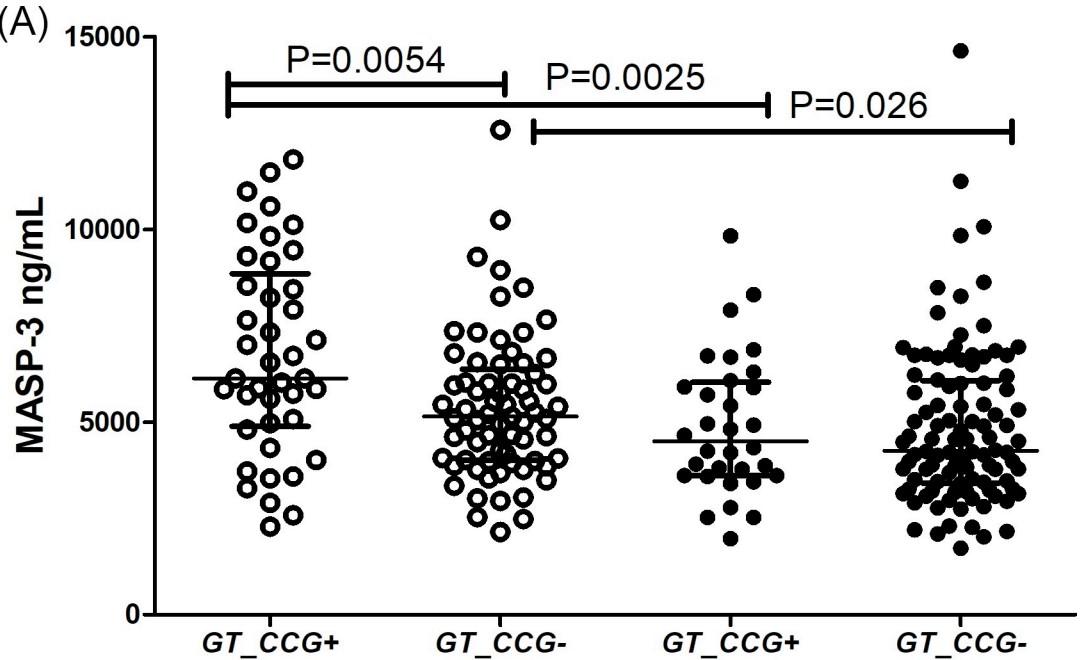

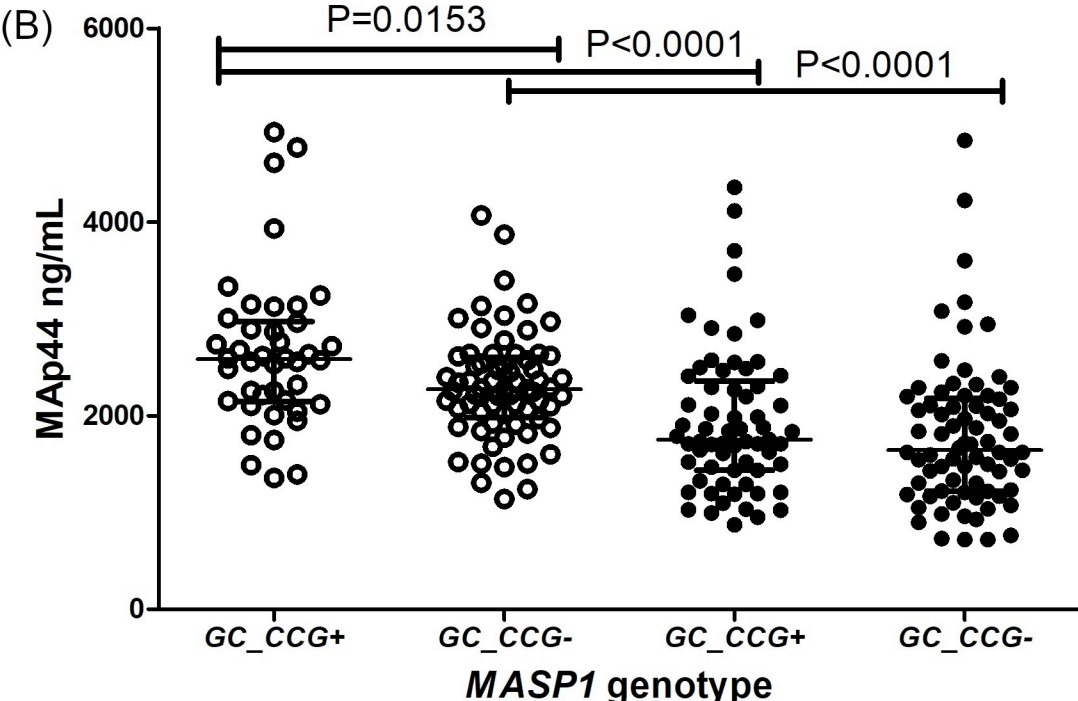

**Fig 5. *MASP1* haplotypes associated with (A) MASP-3 and (B) MAp44 levels.** Data shown with medians and interquartile ranges and Mann-Whitney P values. Open and closed symbols represent controls and patients, respectively. + with the haplotype,—without the haplotype.

MASP-3 protein levels, but only in healthy individuals. The absence of a clear correlation between mRNA levels and protein concentration in serum is not unexpected, since former

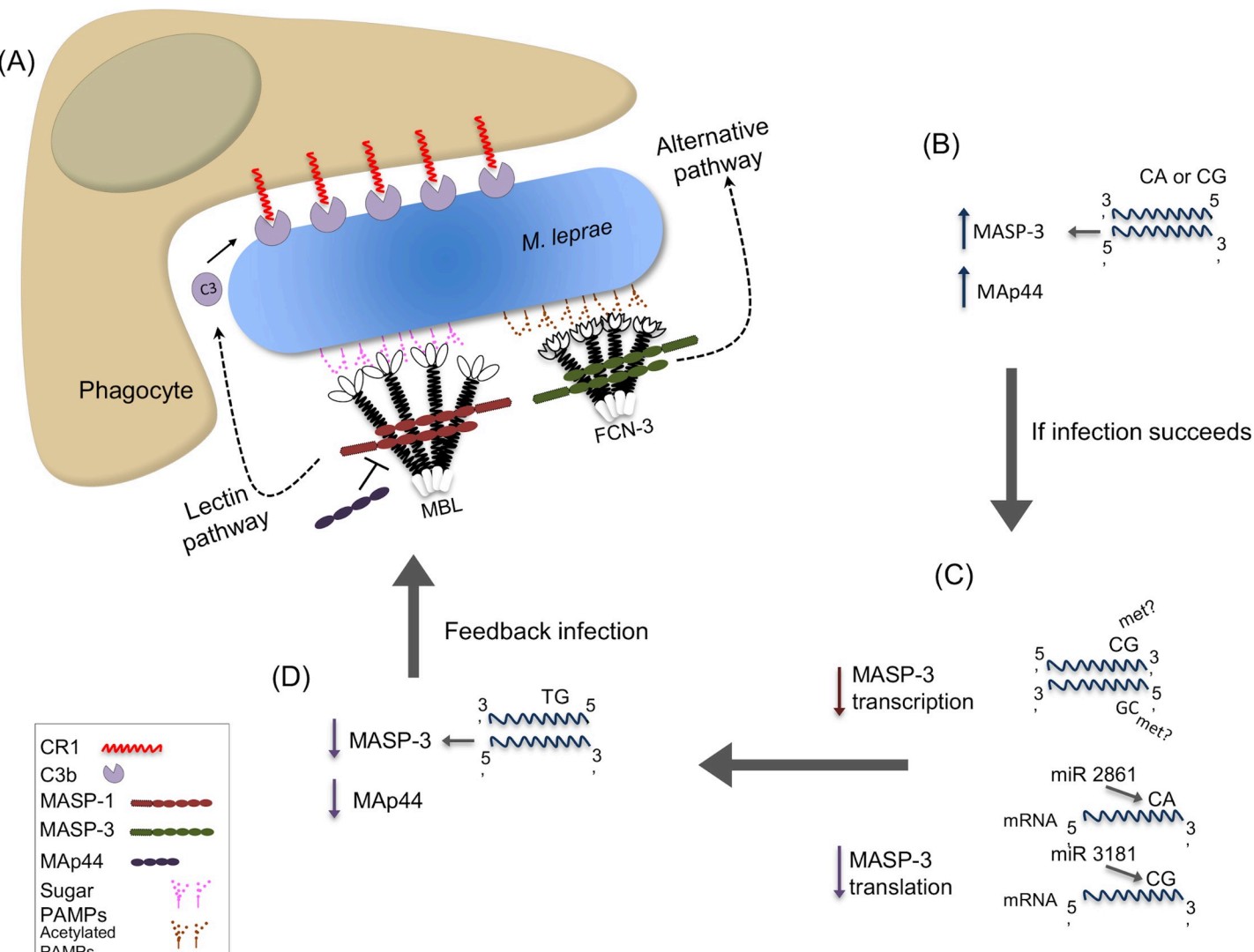

**Fig 6. Proposed roles for *MASP1* products and polymorphisms in susceptibility to *M. leprae* infection. (A)** Collectins (e.g. MBL) or ficolins (e.g. FCN-3) recognize pathogen-associated molecular patterns (PAMPs), composed of sugar/acetylated groups on *M. leprae*. MASP-2 (not depicted in this image) and MASP-1 homodimers complexed with them activate the lectin pathway of complement, whereas MASP-3 may activate the alternative pathway. Both pathways lead to C3b-opsonization and CR1-mediated internalization of the pathogen. **(B)** Healthy individuals with *rs1109452* and *rs850314 CA* or *CG* haplotypes express higher MASP-3 levels. Higher MASP-3 and MAp44 levels were also associated with resistance against the disease. **(C)** CpG methylation at the *CG* haplotype in exon 12 may impair mRNA transcription, spliceosome assembly and mRNA processing. Reduced MASP-3 levels may also result from the differential recognition of *CA* and *CG* haplotypes by miRNAs (miR-2861 and miR-3181, respectively). **(D)** Individuals with *TG* haplotypes present lower baseline MASP-3 levels. Lower MASP-3 and MAp44 levels seem to predispose to the infection, possibly by optimizing opsonin coverage of the parasite.

analyses did not consider different *MASP1* transcripts, and stability of mRNA in cytoplasm may be greatly affected by regulatory mechanisms that were not accounted for in previous transcriptomic analyses.

All exon 12 variants (*rs72549262*G>C*, *rs1109452*C>T* and *rs850314*G>A*) are located within the 3' untranslated region. Those two most downstream (rs1109452 and rs850314) are adjacent to each other, and *CG* represents the most ancestral combination. Thus, the minor alleles *rs1109452*T* and *rs850314*A* disrupt a 5'CpG3' site (where "p" means the phosphodiester bond between *rs1109452*C* and *rs850314*G*). The cytosine of this CpG site was found methylated in the brain [47], but not in cell lines from liver and female reproductive tissue,

where MASP-3 mRNA production is higher (https://gtexportal.org/home/gene/MASP1). DNA methylation in alternatively spliced exons may modulate exon inclusion [48].

Furthermore, the *CA* and *CG* combinations are miRNA targets, as predicted *in silico* using targetScan7.1, and may reduce MASP-3 translation. Thus, one would expect that any nucleotide substitution at these loci would modify gene expression, depending on specific regulatory requirements of the cell type, developmental stage, physiological and immunological responses. In fact, both adjacent polymorphisms were associated with MASP-3 (in the case of rs1109452, even MASP-1) levels. However, the predicted down-regulating effects either of CpG methylation and/or *CA/CG* miRNA binding on MASP-3 levels, were restricted to leprosy patients. In the disease, MASP-3 levels of *CA* or *CG* carriers dropped to the same concentration found in *TG* carriers, who presented the lowest MASP-3 levels, independent of the disease. Interestingly, among the miRNAs predicted to recognize these polymorphic sites, none bind *TG*, but miR-3181 preferentially recognizes *CG* and miR-2861, *CA*. Both are expressed in the liver [49], with miR-2861 being up-regulated by interleukin 6 [50], a proinflammatory cytokine with a pivotal role in leprosy disease [51]. It is thus conceivable that these regulatory mechanisms operate after disease establishment and activation of the acute phase response (Fig 6).

Refining the association analysis to the haplotype level, allowed us to identify the *GT_CCG* and *GC_CCA* haplotypes (containing the previously mentioned *rs850314*A* variant) associated not only with higher MASP-3 levels, but also with higher protection against the disease. Higher MASP-3 levels may avoid initiation of bacterial colonization due to competition with MASP-1 and MASP-2 for binding sites of recognition molecules—blocking the lectin pathway, and/or by competition with binding sites on complement receptors, blocking phagocytosis.

Yet the *GC_CCG* haplotype, associated with leprosy susceptibility, was associated with higher MAp44 serum concentrations. In contrast with MASP-3, however, MAp44 serum levels did not associate with the investigated SNPs, which may suggest other causal variants in linkage disequilibrium with *GC_CCG*, not investigated in this study. In fact, Ammitzboll et al. (2013) [45] list several variants that may modulate MAp44 levels. Furthermore, other factors than those regulating MASP-3 may fit in the present scenario, where MAp44 levels are higher in controls, compared to patients, and in non-lepromatous patients, compared to the more severely affected lepromatous patients.

Beside *GC_CCG*, the haplotypes *GT_CTG* and *GT_GTG* also present at least an additive effect increasing almost twice the susceptibility to the disease. They were not associated with protein levels, although harboring the *rs1109452*T* polymorphism, found associated with higher MASP-1 and lower MAp44 levels. Thus, protein levels shall not be held solely responsible for the association of *MASP1* products with the disease. Beside the pleiotropic nature of the *MASP1* gene itself, the investigated polymorphisms may have effects far beyond those affecting *MASP1*. Other variants linked with those that compose the associated haplotypes, may present epistatic and/or unsuspected pleiotropic effects that affect susceptibility to the disease. In fact, the variants investigated in this study have been recently associated with expression levels of neighboring genes as the ribosomal protein-encoding gene *RPL39L* and the odorant receptor transporters *RTP1*, *RTP3* and *RTP4* (https://gtexportal.org/home/gene/MASP1 and Immun-pop browser). Among them, RTP4 is strongly up-regulated by interferon I, a cytokine known to suppress an adequate cellular response driven by interferon type II against *M. leprae* [52].

In summary, our results show associations of MASP-3 and MAp44 levels with leprosy, as well as with polymorphisms that regulate the levels of these two proteins in serum. Taken together, these results indicate that MASP-3/MAp44 blockage of the lectin pathway may not be the only explanation for resistance to leprosy infection, since expression levels of neighboring genes may be regulated by noncoding polymorphisms investigated in this study. Although

interpreting the evidence is not straightforward, it certainly fosters more investigations on the role played by *MASP1* products in the resistance against mycobacterial infections and its more severe forms. In particular, MASP-3 and MAp44 may be evaluated as new therapeutic agents against leprosy infection and against polarization to lepromatous disease. These observations place the *MASP1* gene and its products in the cross-talk of innate and adaptive immune responses to leprosy infection, its clinical aspects and susceptibility, thus shedding more light on how the complement system may be involved with mycobacterial infections.

## Supporting information

**S1 Fig. Correlations between MASP-1, MASP-3 and MAp44 serum levels in leprosy patients (A-B) and healthy controls (C-D).** Linear regression fit, P and R values are shown. (TIF)

**S1 Table. MASP-1, MASP-3 and MAp44 levels in patients an controls.** n: number of individuals; *: mean protein levels in ug/mL showing: median[IQR] **Levels confering protection against Leprosy infection. Within brackets: minimum and maximal values. [a]: Patients presenting all other forms except Lepromatous and Non-specified (TIF)

**S2 Table. Clinic and demographic characteristics of subjects, MASP1 genotyping and protein levels data used for statistical analyses.** (XLSX)

## Acknowledgments

We deeply thank all patients and controls that volunteered in this study. We also want to thank Sandra J. dos Santos Catarino, Annette G. Hansen and Lisbeth Jensen for all technical guidance and contributions to this work.

## Author Contributions

**Conceptualization:** Hellen Weinschutz Mendes, Angelica Beate Winter Boldt, Ewalda von Rosen Seeling Stahlke, Iara J. Taborda Messias-Reason.

**Data curation:** Hellen Weinschutz Mendes, Angelica Beate Winter Boldt.

**Formal analysis:** Hellen Weinschutz Mendes, Angelica Beate Winter Boldt.

**Funding acquisition:** Jens Christian Jensenius, Steffen Thiel, Iara J. Taborda Messias-Reason.

**Investigation:** Hellen Weinschutz Mendes.

**Methodology:** Hellen Weinschutz Mendes.

**Resources:** Ewalda von Rosen Seeling Stahlke, Jens Christian Jensenius, Steffen Thiel, Iara J. Taborda Messias-Reason.

**Supervision:** Angelica Beate Winter Boldt, Jens Christian Jensenius, Steffen Thiel, Iara J. Taborda Messias-Reason.

**Visualization:** Hellen Weinschutz Mendes.

**Writing – original draft:** Hellen Weinschutz Mendes.

**Writing – review & editing:** Angelica Beate Winter Boldt, Steffen Thiel.

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
