## [Decision Letter · Decision Letter 0]

20 Sep 2019

Dear MSc Weinschutz Mendes:

Thank you very much for submitting your manuscript "Adding MASP1 to the lectin pathway - leprosy association puzzle: hints from gene polymorphisms and protein levels." (#PNTD-D-19-00912) for review by PLOS Neglected Tropical Diseases. Your manuscript was fully evaluated at the editorial level and by independent peer reviewers. The reviewers appreciated the attention to an important problem, but raised some substantial concerns about the manuscript as it currently stands. These issues must be addressed before we would be willing to consider a revised version of your study. We cannot, of course, promise publication at that time.

We therefore ask you to modify the manuscript according to the review recommendations before we can consider your manuscript for acceptance. Your revisions should address the specific points made by each reviewer. 

When you are ready to resubmit, please be prepared to upload the following:

(1) A letter containing a detailed list of your responses to the review comments and a description of the changes you have made in the manuscript.

(2) Two versions of the manuscript: one with either highlights or tracked changes denoting where the text has been changed (uploaded as a "Revised Article with Changes Highlighted" file); the other a clean version (uploaded as the article file).

(3) If available, a striking still image (a new image if one is available or an existing one from within your manuscript). If your manuscript is accepted for publication, this image may be featured on our website. Images should ideally be high resolution, eye-catching, single panel images; where one is available, please use 'add file' at the time of resubmission and select 'striking image' as the file type. 

Please provide a short caption, including credits, uploaded as a separate "Other" file. If your image is from someone other than yourself, please ensure that the artist has read and agreed to the terms and conditions of the Creative Commons Attribution License at http://journals.plos.org/plosntds/s/content-license (NOTE: we cannot publish copyrighted images). 

(4) If applicable, we encourage you to add a list of accession numbers/ID numbers for genes and proteins mentioned in the text (these should be listed as a paragraph at the end of the manuscript). You can supply accession numbers for any database, so long as the database is publicly accessible and stable. Examples include LocusLink and SwissProt.

(5) To enhance the reproducibility of your results, we recommend that you deposit your laboratory protocols in protocols.io, where a protocol can be assigned its own identifier (DOI) such that it can be cited independently in the future. For instructions see http://journals.plos.org/plosntds/s/submission-guidelines#loc-methods

While revising your submission, please upload your figure files to the Preflight Analysis and Conversion Engine (PACE) digital diagnostic tool, https://pacev2.apexcovantage.com/ PACE helps ensure that figures meet PLOS requirements. To use PACE, you must first register as a user. Then, login and navigate to the UPLOAD tab, where you will find detailed instructions on how to use the tool. If you encounter any issues or have any questions when using PACE, please email us at figures@plos.org.

We hope to receive your revised manuscript by Nov 19 2019 11:59PM. If you anticipate any delay in its return, we ask that you let us know the expected resubmission date by replying to this email.

To submit a revision, go to https://www.editorialmanager.com/pntd/ and log in as an Author. You will see a menu item call Submission Needing Revision. You will find your submission record there. 

Sincerely,

Krithivasan Sankaranarayanan, Ph.D.

Guest Editor

Mathieu Picardeau

Deputy Editor

Reviewer's Responses to Questions

**Key Review Criteria Required for Acceptance?**

**Methods**

-Are the objectives of the study clearly articulated with a clear testable hypothesis stated?

-Is the study design appropriate to address the stated objectives?

-Is the population clearly described and appropriate for the hypothesis being tested?

-Is the sample size sufficient to ensure adequate power to address the hypothesis being tested?

-Were correct statistical analysis used to support conclusions?

-Are there concerns about ethical or regulatory requirements being met?

Reviewer #1: Objectives are articulated with the hypothesis to be tested. 

The methods are clearly described.

Reviewer #2: The objectives of the study are clearly presented, and the study design and sample size seem to be adequate to address the hypotheses. The statistical analyses seem to be performed correctly, but this is not my area of expertise, so I would defer to other reviewers' comments for the same. I have no concerns about ethical or regulatory requirements.

Reviewer #3: Please refer to "Summary and General Comments" below.

**Results**

-Does the analysis presented match the analysis plan?

-Are the results clearly and completely presented?

-Are the figures (Tables, Images) of sufficient quality for clarity?

Reviewer #1: The results are presented adequately and respond to the proposed objectives.

Reviewer #2: Overall, the analyses presented match the analysis plan and the results are presented in a clear manner. The figures and tables are okay. However, a couple of clarifications from the authors are necessary:

How do the authors define non-lepromatous, especially w.r.t the numbers given Table 1? Does non-lepromatous include all the patients who were borderline, indeterminate, and tuberculoid?

Can the authors show that the sample sizes within the leprosy cases (for example, 36 non-lepromatous and 97 lepromatous) are sufficient to detect associations with type of leprosy and MASP1 haplotypes/protein levels?

The authors use thresholds for MASP-3 levels of 5,500 ng/mL and for MAp44 of 2,300 ng/mL, respectively, for their analyses; are these clinically determined or relevant thresholds, or just randomly chosen by the authors?

Reviewer #3: Please refer to "Summary and General Comments" below.

**Conclusions**

-Are the conclusions supported by the data presented?

-Are the limitations of analysis clearly described?

-Do the authors discuss how these data can be helpful to advance our understanding of the topic under study?

-Is public health relevance addressed?

Reviewer #1: Authors' conclusions are based on the findings.

Reviewer #2: The conclusions are fairly straightforward, however the concluding paragraph ends rather abruptly. While not strictly necessary, I think the paper would benefit from a few more sentences addressing the public health relevance of this study.

Reviewer #3: Please refer to "Summary and General Comments" below.

**Editorial and Data Presentation Modifications?**

Reviewer #1: - Line 120: there is a repetition of the word "molecules";

- Line 143: repetition of the word "in";

- Methods: the authors could provide information about the study design. Was it a case-control study?

- Table 2: the format does not allow the visualization of all the data;

- Line 528: the word "cornification" could be replaced by classification;

- Line 666: the word "resistance' is not written correctly;

- Figure 6: if possible, improve the resolution for better visualization of the data.

Reviewer #2: Overall, the Introduction is very long-winded and could be reorganized for better flow. In the first paragraph, an explanation of the spectrum of disease manifestation in case of leprosy, would be appreciated. This could then lead to how the complement system modulates the course of the disease towards either pole. The information in Lines 118-139 should be condensed. Again, while I appreciate the thorough introduction to the three MASP1-gene products, the information can be presented in a more concise manner. While talking about the findings of previous studies w.r.t role of complement factors and leprosy susceptibility, the authors should specify whether all the findings are from a particular population (example, the Brazilian population) or from different populations. 

Formatting of the tables needs to be checked to make sure all text is visible and is not cut off (specifically Table 2 and Table 3).

In general, the manuscript needs to be thoroughly checked for grammatical and typographical errors. Some examples:

Line 523 - Change "Mycobacteria" to "mycobacterial"

Line 611 – Reference is missing

Line 665 – Correct spelling of "straightforward"

Line 666 – Correct spelling of "resistance"

Line 641 - Missing in-text citation number for Ammitzboll et al. (2013)

Line 646 - Rephrase "increasing almost twice susceptibility"

Table S1 - The main text mentions 97 lepromatous cases, but Table S1 mentions 98. Please correct the discrepancy. 

The spelling of lepromatous needs to be corrected in the legend. 

The in-text citation style needs to be corrected. Please check the PLoS requirements, which state that "In the text, cite the reference number in square brackets."

Reviewer #3: Please refer to "Summary and General Comments" below.

**Summary and General Comments**

Reviewer #1: The study aims to investigate the role of variants in the MASP1 gene and its products in susceptibility to leprosy. Their results contribute to the understanding of the complexity of the immune response triggered by exposure to Mycobacterium leprae, especially the lectin pathway.

Reviewer #2: In this study, Mendes et al. study the MASP1 gene haplotypes and protein product levels in a cohort of leprosy patients and healthy controls. They show that these haplotypes and protein products, which form part of the complement system, can confer protection/susceptibility to leprosy disease. This seems to be the first study to look at this particular gene and its products in the context of leprosy, as well as in the population under study, although it has been looked at in other mycobacterial diseases such as TB. As such, it adds to our knowledge of the impact of immunogenetics on leprosy disease progression.

Reviewer #3: The authors describe an association study of variants of the MASP1 gene and both leprosy phenotypes and protein expression in a small case-control Brazilian sample. The study is well executed, and results are potentially interesting; however, upon careful reading of the manuscript, a few general issues and a number of specific issues emerge, as follows:

General (major) issues:

1. The writing style adopted is particularly confusing and should be reconsidered. For example, there seems to be no systematic description of the results – the authors seemingly jump from individual marker to 2-marker, 3-marker or 5-marker haplotypic analysis at random, which makes the interpretation of the results very difficult. This also seems to reflect on the key table (table 3, please refer to the next comment) of the study. Also, the manuscript will greatly benefit from a revision by a native English speaker;

2. Tables 2 and 3 are truncated and impossible to read. Please provide readable versions.

Specific (minor) issues:

1. The authors used the Ridley & Jopling leprosy classification system; however, some of the tests necessary for the classic R & J protocol described in the original paper, referenced by the authors (such as the Mitsuda test), are no longer available; how did the authors performed the classification? This is critical given that some of the most interesting results come from comparison involving the clinical forms of disease;

2. In “methods”, the control group is described as composed by blood donors; could the authors clarify how matching for socio-economical and geographic background was achieved?

3. The authors claim that ethnicity of cases and controls was defined based on “ancestry information” but it is not clear how this has been achieved. The sentence “This means 9%...” (lines 215-219) is obscure, please clarify;

4. There is a major difference in age between cases and controls, with the controls being much younger; this could pose a problem for a phenotype such as infection, given that disease risk increases dramatically with age. How does the authors deal with this difference? Shouldn’t this be addressed in the discussion?

5. The strategy for marker selection adopted have limitations well pointed by the authors in the discussion; why didn’t the authors performed complete physical coverage of the candidate gene?

6. Why only a sub-sample was used for the serum concentration assays – in particular, the number of controls is much reduced to almost half (116 out of 214). Is this subsample representative of the total sample described on table 1?

7. The authors describe a multivariate logistic regression analysis including several co-variates obtained in previous studies (pages 305-307)? What was the rationale for such strategy?

8. In “Results”, table 3 (truncated) seems to present data from the 5 markers individually and for the complete 5-markers haplotypes; however, the text describes results for the 2- and 3-markers haplotypes; it would be much less confusing if table 3 includes all genotypic data, with complete info on allele frequencies for all individual markers, 2-, 3- and 5-markes haplotypes;

9. The information conveyed between lines 391-396 – in particular, the sentence ‘In accordance… disease” is confusing, please clarify.

PLOS authors have the option to publish the peer review history of their article (what does this mean?). If published, this will include your full peer review and any attached files.

Reviewer #1: No

Reviewer #2: No

Reviewer #3: No

---

## [Decision Letter · Decision Letter 1]

21 Feb 2020

Dear MSc Weinschutz Mendes,

We are pleased to inform you that your manuscript 'Adding MASP1 to the lectin pathway - leprosy association puzzle: hints from gene polymorphisms and protein levels.' has been provisionally accepted for publication in PLOS Neglected Tropical Diseases.

Before your manuscript can be formally accepted you will need to complete some formatting changes, which you will receive in a follow up email. A member of our team will be in touch within two working days with a set of requests.

Best regards,

Mathieu Picardeau

Deputy Editor

Reviewer's Responses to Questions

**Key Review Criteria Required for Acceptance?**

**Methods**

-Are the objectives of the study clearly articulated with a clear testable hypothesis stated?

-Is the study design appropriate to address the stated objectives?

-Is the population clearly described and appropriate for the hypothesis being tested?

-Is the sample size sufficient to ensure adequate power to address the hypothesis being tested?

-Were correct statistical analysis used to support conclusions?

-Are there concerns about ethical or regulatory requirements being met?

Reviewer #1: Yes.

Reviewer #2: (No Response)

Reviewer #3: (No Response)

**Results**

-Does the analysis presented match the analysis plan?

-Are the results clearly and completely presented?

-Are the figures (Tables, Images) of sufficient quality for clarity?

Reviewer #1: Yes.

Reviewer #2: (No Response)

Reviewer #3: (No Response)

**Conclusions**

-Are the conclusions supported by the data presented?

-Are the limitations of analysis clearly described?

-Do the authors discuss how these data can be helpful to advance our understanding of the topic under study?

-Is public health relevance addressed?

Reviewer #1: Yes.

Reviewer #2: (No Response)

Reviewer #3: (No Response)

**Editorial and Data Presentation Modifications?**

Reviewer #1: (No Response)

Reviewer #2: Line 97 – To this reviewer, it is not clear what the "cc" stands for.

Line 107 – Ref 39 needs to be added properly.

Line 111 – Use proper in-text citation for West and Kemper

Lines 351 and 368 – Change to P <0.0004 and P<0.0001, respectively, for uniform use of “.” as decimal separator

Reviewer #3: (No Response)

**Summary and General Comments**

Reviewer #1: In the first review, I suggested some minor corrections. In the updated version of the paper, the authors answered the questions raised by me and the other reviewers, which I believe has improved the quality and clarity of the manuscript.

I would like to add just one suggestion. In defining the study design, I suggest that the authors change the second design to "cross-sectional study", replacing the term transversal epidemiological.

Also, it was not clear what cc. means (line 97 - introduction).

Reviewer #2: The authors have sufficiently addressed my concerns/comments on the previous draft of the manuscript. The overall flow of the manuscript has improved significantly. A few minor formatting corrections (see Editorial and Data Presentation Modifications) need to be made. Assuming these are made, I recommend the article be accepted for publication.

Reviewer #3: The authors addressed adequately all the concerns raised by this reviewer and in my opinion, the manuscript is suitable for publication.

PLOS authors have the option to publish the peer review history of their article (what does this mean?). If published, this will include your full peer review and any attached files.

Reviewer #1: No

Reviewer #2: No

Reviewer #3: No

---

## [Editor Report · Acceptance letter]

20 Mar 2020

Dear MSc Weinschutz Mendes,

We are delighted to inform you that your manuscript, "Adding MASP1 to the lectin pathway - leprosy association puzzle: hints from gene polymorphisms and protein levels.," has been formally accepted for publication in PLOS Neglected Tropical Diseases.

Best regards,

Serap Aksoy

Editor-in-Chief

Shaden Kamhawi

Editor-in-Chief
